# OpenReview forum: "Lifelong Unlearning for Multimodal Large Language Models"
_ICLR.cc/2026/Conference — Submitted to ICLR 2026_

### Official Review · Reviewer_cC9S · 2025-10-26

**Soundness:** 2
**Presentation:** 2
**Contribution:** 3
**Rating:** 4
**Confidence:** 4

**Summary:**

They propose MLUBench. It evaluates the performance of unlearning in a lifelong setting, where multiple unlearning requests happen sequentially. They reveal that typical unlearning approaches (GA, NPO etc) fail in this setting, and propose a MoE-inspired method called LUMoE, which leverages switchable LoRA adapters, each of which is responsible for a refusal against the corresponding task, achieving high forget quality without utility degradation.

**Strengths:**

1. **Extensive benchmark coverage**: Although comparison with related works is insufficient (detailed in weakness), their proposed benchmark is pretty extensive in terms of size and domain coverage.

2. **Diverse analysis**: The authors conduct extensive analysis ranging from adversarial robustness and key component ablations.

3. **Reasonable task design**: Evaluation of “lifelong” unlearning is practically important. I think it is also practical that they aim at evaluating the unlearning of pretrained knowledge (instead of the more common yet easier finetuned knowledge). Their definition of forget quality (focusing on refusal and not counting hallucinating answers as a success, described in Section 6.1.1) is reasonable as well.

**Weaknesses:**

Despite their benchmark contributions seeming meaningful, I find myself slightly doubting them due to their inadequate presentation.

1. **Limited related work coverage on multimodal unlearning** (Section 2, line 112-121): First, the authors compare their proposed MLUBench with MMUBench (and FIUBench in the appendix), but comparisons with other recent ones, such as MLLMU-Bench [1], are missing. Second, they argue as if Liu et al.’s SIU is the only method, but I am at least aware of MMUnlearner [2] as another method. Third, PULSE [3] reports a similar performance drop in sequential unlearning; the authors can clarify how their contribution relates to it. In summary, the related work survey feels incomplete. Survey papers (e.g., [4]) and some "awesome-unlearning" GitHub repos offer a more extensive coverage of multimodal unlearning works that the authors could build on.

2. **License concern** (Section 4.1, line 210): The authors mention that they crawled images from Google Images. I like its wide domain coverage, including e.g., cartoons, but did the authors confirm that the images used have no license issues? I suspect this has been one of the bottlenecks in creating extensive benchmarks (i.e., realistic unlearning target domains are protected by copyright), so I encourage them to elaborate on it.

3. **Mismatch between evaluation scheme and baseline** (Table 1): Based on their task design, a hallucinating answer may not be considered as a successful unlearning (line 329-330). To achieve a high unlearning score in such a design, LUMoE conducts preference optimization w.r.t. the refusal response set (Section 5.2 and Table 19).  I like this design itself, but if the authors go in this direction, the baseline approach should be some DPO or PO-based approach (perhaps with some regularizer to retain its utility) that explicitly guides the model to refuse forget set queries. I don’t doubt the effectiveness of their proposed LUMoE approach, but GA, NPO, etc, (which only decrease the likelihood of correct output without an explicit positive guidance toward refusal) have an inevitable issue of hallucination, and are not the best baseline to compare against.

4. **Generalizability beyond Llava evaluation** (Section 4.1, line 238): The content of the MLUBench is selected such that LLaVA models can answer them perfectly, enabling an unlearning efficiency measured from an initial 100% accuracy if (and perhaps only if) users want to evaluate LLaVA. Do other LMMs achieve similarly high pre-unlearning accuracy on MLUBench before unlearning? If not, how should performance be compared across models?

5. **Limited information in ablation** (Section 6.4): Firstly, why is LUMoE not evaluated on TruthfulQA (Appendix D.6)? I think currently it is unclear whether LUMoE can preserve general utility after sequential unlearning. Secondly, while they perform a pretty extensive ablation, most of the results are fully deferred to the appendix, only mentioning e.g., “it does not affect our primary conclusion (line 463)”. While I believe the authors did it mainly because of the space constraint, this makes me feel they are hiding something in the appendix. I would expect to have at least one sentence of quantitative argument for each of the discussions within the main paper.

[1] Liu et al, Protecting Privacy in Multimodal Large Language Models with MLLMU-Bench

[2] Huo et al, MMUnlearner: Reformulating Multimodal Machine Unlearning in the Era of Multimodal Large Language Models

[3] Kawakami et al, PULSE: Practical Evaluation Scenarios for Large Multimodal Model Unlearning

[4] Feng et al, A Survey on Generative Model Unlearning: Fundamentals, Taxonomy, Evaluation, and Future Direction

**Questions:**

I list up the points that could further improve their presentation. However, these points are relatively minor. Therefore, I would urge the authors to address the points listed in weaknesses first, which could potentially affect their claimed contribution significantly.

1. **Domain-stratified analysis** (Section 6.4): A key claimed contribution is broad domain coverage, but the paper doesn’t analyze whether this breadth yields insights that existing face-focused benchmarks would miss. The only discussion I find somewhat related is the task-ordering ablation (lines 455-459), where the authors concluded that it does not affect the unlearning efficiency. Are there any phenomena that are observable only in some specific domains?

2. **Slightly more sophisticated adversarial attacks** (Section 6.4): Currently, they confirm that the AutoDAN attack does not work. This discussion is useful given the increasing amount of work in adversarial unlearning. However, since LUMoE aims at routing to a refusal path (and not fundamentally removing the knowledge), there should be a more diverse attack vector. In particular, LUMoE may be vulnerable if the routing module (Section 5.2) misroutes and the query is handled by the base model. Can adversaries craft prompts that trigger routing failures and force processing by the base model?

**Details Of Ethics Concerns:**

As I pointed out as a weakness, the authors mention that they crawled images from Google Images (Section 4.1, line 210), but there is no justification for whether they verified the licenses of the crawled images.

---

> ### Author Response · Authors · 2025-11-20
> **Response to reviewer cC9S**
>
> # Response to reviewer cC9S (1/2)
>
> We are glad that the reviewer found the proposed benchmark extensive. We would like to respond to each point individually.
>
> > R1: Limited related work coverage on multimodal unlearning
>
> We thank the reviewer for the suggestions on the related work coverage. Following your advice, we have **cited and discussed** these excellent works in the **related work**.
>
> **Machine Unlearning for Language Models**
>
> With respect to the MLLMs unlearning, Liu et al. [1] propose the MLLMU-Bench, a comprehensive benchmark targeted at multimodal unlearning, which includes various multimodal profiles. MMUNLEARNER [2] is a novel geometry-constrained gradient ascent method designed for MLLMs unlearning. It can effectively preserve MLLMs' overall abilities. In addition, Feng et al. [4] perform a systematic review of the generative model unlearning, including multimodal unlearning. This comprehensive survey covers various multimodal unlearning methods, evaluation metrics, and open challenges.
>
> Compared with the MLLMU-Bench, **our dataset covers a broader type of unlearning entities other than the celebrities**. Therefore, we demonstrate the failures of current methods on a broader scale. In addition, our method differs from MMUNLEARNER, which our **LUMoE targets at MLLM lifelong unlearning rather than MLLM unlearning**.
>
>
> **Sequential Unlearning for Language Models**
>
> Kawakami et al. [3] explore the evaluation framework for the unlearning of large multimodal models (LMMs). They introduce the advanced PULSE protocol for two essential applications: the unlearning of pre-trained knowledge and the sustainability evaluation. They conduct extensive experiments and reveal the critical limitations of current unlearning methods.
>
> Our work strongly supports the findings of PULSE with **unique contributions**. Specifically, we introduce **a new and expensive benchmark** and demonstrate the failures of current methods **on a broad scale**. In addition, we propose LUMoE, **a novel method designed to solve the challenges of lifelong unlearning**. In a nutshell, PULSE proves the necessity of a new approach, and our work provides both a rigorous demonstration of this need and the novel solution (LUMoE).
>
> [1] Liu et al, Protecting Privacy in Multimodal Large Language Models with MLLMU-Bench
>
> [2] Huo et al, MMUnlearner: Reformulating Multimodal Machine Unlearning in the Era of Multimodal Large Language Models
>
> [3] Kawakami et al, PULSE: Practical Evaluation Scenarios for Large Multimodal Model Unlearning
>
> [4] Feng et al, A Survey on Generative Model Unlearning: Fundamentals, Taxonomy, Evaluation, and Future Direction
>
> > R2: License concern
>
> We thank the reviewer for raising this important point regarding image licenses. Our use of images crawled from Google Images is strictly for **non-commercial, academic research**, which we believe falls under the [fair use principles](https://www.copyright.gov/fair-use). In response to this comment, we have added a formal **copyright disclaimer** at the **beginning of Section 4** to explicitly address this. The disclaimer affirms that all images are used for **scholarly analysis**, do not impact the market value of the original works, and that **all rights remain with their respective copyright holders**.
>
> **Copyright Disclaimer**
>
> Images used in this study were collected from publicly available sources via Google Images. In accordance with the [fair use principles](https://www.copyright.gov/fair-use), the use of these images is constrained within scholarly analysis and does not affect the market value of the original works. All rights remain with the original copyright holders.
>
> > R3: Mismatch between evaluation scheme and baseline
>
> We thank the reviewer for this insightful observation. We agree that comparing against a preference optimization (PO) baseline is crucial. To that end, we have already conducted this exact experiment.
>
> As analyzed in **Section 6.4 ("Ablation on the gate module")**, we evaluated a PO-based method [5] by removing our router. The results in **Table 3** of the main text confirm that while PO successfully encourages refusal, it does so indiscriminately, causing a significant utility drop on the Retain Set. This validates the reviewer's point and **underscores the advantage of our LUMoE approach**, which uses selective expert routing to achieve refusal without compromising general performance.
>
> [5] Maini P, Feng Z, Schwarzschild A, et al. TOFU: A Task of Fictitious Unlearning for LLMs[C]//First Conference on Language Modeling.

---

> ### Author Response · Authors · 2025-11-20
> **Response to reviewer cC9S**
>
> # Response to reviewer cC9S (2/2)
>
> > R4: Generalizability beyond Llava evaluation
>
> We thank the reviewer for the suggestion on the pre-unlearning efficiency evaluation. Following your advice, we evaluate other LMMs, including **Qwen2.5-VL-32B-Instruct and Qwen2.5-VL-72B-Instruct** on the MLUBench.
>
> Table: Initial accuracy of Qwen2.5-VL-Instruct series on the MLUBench. All values are expressed as percentages (%).
> | Model | Task A  | Task B  |Task C  | Task D | Overall |
> | --- | --- | --- | --- | --- | --- |
> | Qwen2.5-VL-32B-Instruct | 95 | 84 | 82 | 81 | 92 |
> | Qwen2.5-VL-72B-Instruct | 99 | 96 | 94 | 99 | 99 |
>
> We update the results in the **Pre-Unlearning Accuracy Evaluation** in the **Appendix** of our submission. According to the results, the Qwen2.5-VL-32B-Instruct and Qwen2.5-VL-72B-Instruct also achieve **almost 100% pre-unlearning accuracy** on the MLUBench. Therefore, our benchmark can achieve high pre-unlearning accuracy across different model series.
>
> By the way, if users want to evaluate different model series without a pre-test. They can perform the **supervised fine-tuning using our benchmark**, which can also guarantee high pre-unlearning accuracy.
>
> > R5: Limited information in ablation
>
> We appreciate the suggestion of evaluating the LUMoE's ability to preserve general utility. Following your advice, we **test the LUMoE on the TruthfulQA and MMBench** after the whole unlearning series. The detailed results are updated in the **General-purpose Benchmark Evaluation** in the **Appendix**.
>
> Table: Performance comparison of LLaVA-7B before and after LUMoE.
> | Benchmark | Original Model (%) | After LUMoE (%) |
> | --- | --- | --- |
> | TruthfulQA | 41.25 | 40.75 |
> | MMBench-DEV-EN | 75.77 | 75.42 |
> |MMBench-DEV-CN| 71.59 | 71.52 |
> |CCBench-DEV| 41.42 | 41.23 |
>
> Table: Performance comparison of LLaVA-13B before and after LUMoE.
> | Benchmark | Original Model (%) | After LUMoE (%) |
> | --- | --- | --- |
> | TruthfulQA | 41.25 | 41.00 |
> | MMBench-DEV-EN | 77.39 | 77.25 |
> |MMBench-DEV-CN| 74.29 | 74.06 |
> |CCBench-DEV| 43.38 | 43.33 |
>
> Note: CCBench is a part of the [MMBench](https://github.com/open-compass/MMBench/blob/main/README.md).
>
> The above tables show that our method can **effectively preserve the model's overall abilities.**
>
> In addition, we thank the reviewer for understanding the space constraint. Following your advice, we **update the summarized quantitative argument sentence** to the main text.
>
> > R6: Domain-stratified analysis
>
> We thank the reviewer for raising this important point about domain-stratified analysis. The reviewer is correct that a key contribution is broad domain coverage, and we agree that investigating domain-specific phenomena is a compelling research direction.
>
> In this paper, **our primary goal was to evaluate the general challenges and propose solutions for MLLM lifelong unlearning** across a diverse landscape, which is why our analysis focused on aggregate performance. The task-ordering ablation (Section 6.4) was one example of a general property we tested. A full stratification by domain, while insightful, would be an extensive study in itself and is beyond the scope of our current contribution.
>
> We have designed MLUBench to facilitate such deeper analysis. As a clear next step, researchers (including ourselves) can now isolate domains like "personage" or "movies" to uncover unique unlearning behaviors. We will explicitly note this as a key direction for future work.
>
> > R7: Slightly more sophisticated adversarial attacks
>
> We thank the reviewer for this insightful observation. The reviewer is correct that adversarial prompts designed to cause routing failures present a theoretically viable attack vector. However, in our current framework, the routing modules are **closed-source models** accessed via API, which presents a significant practical barrier to such white-box adversarial attacks. Investigating these more sophisticated attacks on the routing logic is a critical and valuable direction for future work.
>
> If you have any further concerns or questions, feel free to let us know. We are glad to discuss and clarify them in more detail.

---

> > ### Author Response · Authors · 2025-11-25
> >
> > Dear reviewer cC9S,
> >
> > Thanks for taking the time to review our work. We have carefully considered your comments and made every effort to respond to your concerns.
> >
> > If you have any further questions or require additional clarification, please let us know.
> >
> > Best regards,
> >
> > Authors

---

> ### Comment · Reviewer_cC9S · 2025-11-25
> **Response to the Authors**
>
> I appreciate the author’s thorough response. Many of my concerns are resolved, but I think a few high-level concerns remain.
>
> ---
> ✅R1: Now the coverage of related works seems good.
>
> ( ✅ )R2: I think they have now clarified their stance regarding copyright, and I do not see a fatal issue, although I cannot make a rigorous judgment about the validity of their argument, as I am not perfectly familiar with it.
>
> ✅R3: I apologize for missing this table. Now I do not consider this a limitation.
>
> **R4:** The new table with Qwen is good. One thing that slightly triggered me about their response is that the authors seem to mix the unlearning of finetuned knowledge with that of pre-trained knowledge. This ignores some works that separately handle these two paradigms [1-2], and obscures why the authors filtered the data such that Llava can answer perfectly, and therefore I need to partially retract my Strength 3. At a minimum, I think they should make readers aware of the difference if they encourage users to use the benchmark in both paradigms.
>
> ✅R5: This table looks good. Now it is quantitatively visible that their proposed approach maintains utility as expected.
>
> ✅R6: The authors are persuasive at this point, and I think leaving the deep analysis for future work is justifiable to some degree.
>
> **R7:** My point here was not limited to white-box attacks. Let’s say Animals is a forget set, and an adversary aims to extract some information about an animal. Can’t the adversary craft a prompt, which is in reality about an animal, but looks like it does not belong to the Animal class? In a concrete example, *“What is the common name of this animal? Also, tell me about {a bunch of questions unrelated to the animal}”* could force the prompt to be classified to a non-animal class or as “None”. If I understand LUMoE correctly, if such malicious prompting is possible, this query is not handled by the refusing module, and the model leaks the information about the animal.
>
> I think rigorous adversarial robustness of LUMoE can be out of scope, especially because their benchmark proposal alone is already reasonably meaningful, but I am somewhat concerned that the current draft and the authors’ stance during rebuttal might be giving a stronger impression of security than is warranted.
>
> ---
> [1] Kawakami et al, PULSE: Practical Evaluation Scenarios for Large Multimodal Model Unlearning, NeurIPS Workshop 2025.
>
> [2] Yao et al, Machine Unlearning of Pre-trained Large Language Models, ACL2024.

---

> > ### Author Response · Authors · 2025-11-28
> >
> > # Thanks for your reply!
> >
> > Dear reviewer cC9S,
> >
> > We appreciate your engagement with our discussion. We will address your remaining concerns regarding the R4 and R7.
> >
> > > R4: The new table with Qwen is good. One thing that slightly triggered me about their response is that the authors seem to mix the unlearning of finetuned knowledge with that of pre-trained knowledge. This ignores some works that separately handle these two paradigms [1-2], and obscures why the authors filtered the data such that Llava can answer perfectly, and therefore I need to partially retract my Strength 3. At a minimum, I think they should make readers aware of the difference if they encourage users to use the benchmark in both paradigms.
> >
> > Thanks for raising this critical point of the distinction between the unlearning of pre-trained knowledge and the fine-tuned knowledge. We agree that they are different paradigms that should be separated.
> >
> > We would like to clarify that our reference to Supervised Fine-Tuning (SFT) in the previous rebuttal **is not intended to mix the two paradigms.** Our purpose is to demonstrate the generality of our dataset, which can be used to evaluate the unlearning of both knowledge acquired through fine-tuning and knowledge inherent in the pre-trained model.
> >
> > **Following your advice, to avoid the mixture of these two paradigms, we have included an individual section in the Appendix to elaborate on this.**
> >
> > **Pre-trained verse Fine-tuned Unlearning Paradigm**
> >
> > * **Pre-trained knowledge unlearning:** This paradigm mainly targets the models that are equipped with high pre-unlearning accuracy on MLUBench. Therefore, we can consider the MLUBench as the pre-trained knowledge. We mainly performed this paradigm in our main experiments.
> > * **Fine-tuned Knowledge Unlearning:** This paradigm mainly targets the models that are not equipped with high pre-unlearning accuracy on MLUBench. In this paradigm, we can perform the supervised fine-tuning using MLUBench to establish a high pre-unlearning accuracy.
> >
> > **Our current design is in line with the "pre-trained knowledge unlearning" paradigm**. Specifically, the selected entities and questions are those that the LLaVA and Qwen-2.5-VL can answer with almost perfect accuracy before unlearning.
> >
> > However, **our MLUBench is also compatible with the "fine-tuned knowledge unlearning paradigm"**. That is, if a model's pre-unlearning accuracy is not good enough, users can also fine-tune this model on MLUBench to acquire a high pre-unlearning accuracy and evaluate their methods.

---

> > > ### Author Response · Authors · 2025-11-28
> > >
> > > > R7: My point here was not limited to white-box attacks. Let's say Animals is a forget set, and an adversary aims to extract some information about an animal. Can't the adversary craft a prompt, which is in reality about an animal, but looks like it does not belong to the Animal class? In a concrete example, "What is the common name of this animal? Also, tell me about {a bunch of questions unrelated to the animal}" could force the prompt to be classified to a non-animal class or as "None". If I understand LUMoE correctly, if such malicious prompting is possible, this query is not handled by the refusing module, and the model leaks the information about the animal. I think rigorous adversarial robustness of LUMoE can be out of scope, especially because their benchmark proposal alone is already reasonably meaningful, but I am somewhat concerned that the current draft and the authors' stance during rebuttal might be giving a stronger impression of security than is warranted.
> > >
> > > Thank the reviewer for pointing out the overclaiming problem regarding the adversarial robustness. We agree that LUMoE cannot fully guarantee adversarial robustness. Following your advice, we have **weakened our claim about the adversarial robustness.**
> > >
> > > We have **added the following discussion in the limitation section in the Appendix.**
> > >
> > > **Limitations and Future Directions**
> > > While we have validated the effectiveness of LUMoE under the jailbreak attack. Our proposed method cannot fully guarantee resistance to the sophisticated adversarial attacks. There exists a partial leakage risk when the attacks are directly targeting the router. However, we view this as an important future work and **propose potential future solutions**:
> > > * **Detect adversarial prompts:** We can add a detection module for adversarial attacks. An option is to detect the misleading prompts like  "tell me about {a bunch of questions unrelated to the animal}". When such an attack is detected, the router can move the input to the unlearning adapter.
> > > * **Second round filtration for "None":** We can revise the default setting so that the "None" ** will not trigger the input into the base model**. When the router outputs "None" and decides to direct the input into the original model. We can employ another light-weight auditor (e.g., a classifier) to perform a second round detection of whether the input contains entities that should be forgotten. If it does, the router should move the input to the unlearning adapter.
> > >
> > > As we emphasized in our previous rebuttal, adversarial robustness is beyond the scope of our current research.
> > >
> > > We hope our clarifications and revisions can address your remaining concerns.
> > >
> > >
> > > Best regards,
> > >
> > > Authors

---

### Official Review · Reviewer_wX2X · 2025-10-29

**Soundness:** 2
**Presentation:** 3
**Contribution:** 2
**Rating:** 4
**Confidence:** 4

**Summary:**

This paper addresses the lifelong unlearning problem in Multimodal Large Language Models (MLLMs). To investigate this issue, this paper constructs MLUBench, a new benchmark consisting of 5,105 images and 15,414 Visual Question Answering (VQA) pairs. To solve this task, this paper proposes LUMoE, a novel unlearning method based on the LoRA modules. Experimental results demonstrate that LUMoE significantly outperforms existing approaches, achieving superior performance across multiple evaluation settings.

**Strengths:**

- S1. This problem setting, which addresses continual unlearning requests on MLLMs, is both practical and important, as it aligns with real-world needs for handling sequential unlearning scenarios.

- S2. A major limitation in this research area is that existing benchmarks often contain only a small number of tasks and are restricted to specific domains, such as facial images. Consequently, the construction of a large-scale VQA dataset in this study represents a certain contribution to advancing research on unlearning in LMMs.

**Weaknesses:**

- W1. **Lack of the mention of similar work**.  Although this paper states that it is the first to address continual unlearning for MLLMs, similar ideas have been explored in prior research, such as [1]. It would be helpful if this paper could acknowledge and discuss this related work to better position its novelty and contribution within the existing literature.

- W2. **Lack of general-purpose benchmark evaluation.** Although the retain set and forget set are both sampled from the same VQA pool, evaluating performance only on this pool may not be sufficient to support the claim that the model’s overall capability is preserved. It would be valuable for this paper to additionally evaluate the model on a general-purpose benchmark, such as MMBench, by excluding questions similar to those in the forget set. This would provide stronger evidence that the model maintains its generalization ability.

- W3. **The proposed method appears somewhat naive, and not practical.** This paper provides only a shallow analysis of why existing approaches fail, so the proposed LUMoE is the main contribution. However, to my understanding, the proposed approach essentially trains a separate LoRA for each unlearning task and employs a powerful LLM (GLM-4V-Plus) for routing among them. While this design may yield strong empirical results, it also introduces significant operational overhead, since a new set of LoRA parameters must be maintained for each unlearning request, and the routing model depends on a large, resource-intensive LLM. These factors raise concerns about the scalability and practicality of the approach, and the overall method still feels too naive in its design.

- W4. **A deeper analysis would be desirable.** The method proposed in this paper appears to be only a naive solution to the problem. It would be more valuable if the paper provided a deeper investigation into why existing methods fail: for example, a detailed analysis of whether the failure of unlearning stems from the Vision side or the LLM side, or how performance changes when the training parameter (etc., projector, LoRA, full-tuning) is altered. Designing a proposed method built on such an analysis would make this paper much more meaningful.


From these perspectives, I would give this paper a Borderline Reject rating.

[1] Kawakami+, PULSE: Practical Evaluation Scenarios for Large Multimodal Model Unlearning, NeurIPS Workshop 2025

**Questions:**

- Q1. I ask the authors to emphasize the importance of the LUMoE method from both a practical and an academic perspective.

- Q2. I would like to recommend that the authors add the evaluation results on general-purpose benchmarks.

---

> ### Author Response · Authors · 2025-11-20
> **Response to reviewer wX2X**
>
> # Response to reviewer wX2X (1/2)
>
> We are glad that the reviewer found the studied problem practical and important. We would like to respond to each detailed point individually.
>
> > R1: W1. Lack of the mention of similar work.
>
> We appreciate the reviewer's suggestion on the related works. Following your advice, we have **cited and thoroughly discussed** [1] in the **related work** of our submission.
>
> **Sequential Unlearning for Language Models**
>
> Kawakami et al. [1] explore the evaluation framework for the unlearning of large multimodal models (LMMs). They introduce the advanced PULSE protocol for two essential applications: the unlearning of pre-trained knowledge and the sustainability evaluation. They conduct extensive experiments and reveal the critical limitations of current unlearning methods.
>
> Our work strongly supports the findings of PULSE with **unique contributions**. Specifically, we introduce **a new and expensive benchmark** and demonstrate the failures of current methods on **a broad scale**. In addition, we propose LUMoE, **a novel method designed to solve the challenges of lifelong unlearning**. In a nutshell, PULSE proves the necessity of a new approach, and our work provides both a rigorous demonstration of this need and the novel solution (LUMoE).
>
> [1] Kawakami+, PULSE: Practical Evaluation Scenarios for Large Multimodal Model Unlearning, NeurIPS Workshop 2025
>
>
> > R2: W2. Lack of general-purpose benchmark evaluation.
>
> We thank the reviewer for the insightful suggestion regarding general-purpose benchmark evaluation. To directly address this point and provide stronger evidence for preserved generalization, we have **evaluated our method on TruthfulQA and MMBench**.  The results, summarized in the following tables, show that **the performance drop is negligible (less than 0.6% in both cases)**, indicating that LUMoE successfully retains the model's overall capabilities. The detailed results are updated in the **General-purpose Benchmark Evaluation** in the **Appendix** of our submission.
>
> Table: Performance comparison of LLaVA-7B before and after LUMoE.
> | Benchmark | Original Model (%) | After LUMoE (%) |
> | --- | --- | --- |
> | TruthfulQA | 41.25 | 40.75 |
> | MMBench-DEV-EN | 75.77 | 75.42 |
> |MMBench-DEV-CN| 71.59 | 71.52 |
> |CCBench-DEV| 41.42 | 41.23 |
>
> Table: Performance comparison of LLaVA-13B before and after LUMoE.
> | Benchmark | Original Model (%) | After LUMoE (%) |
> | --- | --- | --- |
> | TruthfulQA | 41.25 | 41.00 |
> | MMBench-DEV-EN | 77.39 | 77.25 |
> |MMBench-DEV-CN| 74.29 | 74.06 |
> |CCBench-DEV| 43.38 | 43.33 |
>
> Note: CCBench is a part of the [MMBench](https://github.com/open-compass/MMBench/blob/main/README.md).
>
>
> > R3: W3. The proposed method appears somewhat naive, and not practical.
>
> We thank the reviewer for raising the critical points of practicality and novelty. We would like to emphasize the importance of our method from both the practical and the academic novelty perspectives.
>
> **1. Academic Novelty**
>
> We respectfully argue that the novelty of LUMoE lies not in inventing new components, but in their **novel integration and adaptation to solve the uniquely challenging problem of MLLM lifelong unlearning**. We identify the key challenges as the continual destruction of unlearning methods from both the LLMs and the vision sides. Therefore, inspired by this analysis, we propose LUMoE to isolate each unlearning edit into a modular LoRA adapter. While we acknowledge that the MoE and LoRA are established techniques, applying them to the lifelong unlearning for MLLMs is still an unexplored task. Therefore, on the academic novelty, LUMoE's contribution is algorithmic and architectural.
>
> **2. Practicality and Scalability**
>
> We understand the reviewer's concerns over the practicality and scalability. However, we respectfully argue that the cost of our operations is **highly manageable**.
>
> - **Computational Cost:** The main operations include the training of LoRA adapters and routing. As shown in the **Table 5** of the main text, training one adapter is highly efficient, **averaging 11 minutes**. The routing operation takes **4 seconds** on average. In addition, we employ the routing model through **API calling**, which effectively offloads the local computing cost.
> - **Storage Cost:** In our experiments, each adapter is about **170MB**. In the context of modern storage solutions, such a cost may be insignificant. This is particularly true when weighed against the degradation of a multi-billion-parameter model, which may need expensive retraining.

---

> ### Author Response · Authors · 2025-11-20
> **Response to reviewer wX2X**
>
> # Response to reviewer wX2X (2/2)
>
> > R4: W4. A deeper analysis would be desirable.
>
> We thank the reviewer for the suggestions on the deeper analysis. Following your advice, we have provided additional **deep analysis** in the **Appendix** of our submission.
>
> **Current Methods**
>
> Current unlearning methods mainly perform destructive weight updating on models. This includes gradient-based methods (GA, GD, KL) and alignment-based ones (NPO). Under the continual unlearning setting, such destructive effects may accumulate.
>
> **A Deeper Analysis**
>
> We believe the failure of existing methods may stem from the continuous destruction of the knowledge of **both the Vision and LLM sides**.
>
> - **On the LLM side:** Continual unlearning continuously corrupts the LLM's weights. Since the knowledge in LLMs may be entangled, the unlearning operation may undermine LLMs' overall abilities when erasing target knowledge.
> - **On the vision side:** Continual unlearning continuously alters the vision adapter to forget specific objects, which may degrade its general feature adaptation capabilities for untargeted objects.
> - **On the multimodal alignment side:** In addition, the alignment between vision and language may break down when vision representations are continuously perturbed.
>
> This analysis inspired our design for LUMoE. LUMoE assigns each unlearning task to its own separate adapter. Crucially, the base models (Vision and LLM) remain frozen. This approach directly avoids both cumulative damage and representational drift.
>
> > R5: Q1. I ask the authors to emphasize the importance of the LUMoE method from both a practical and an academic perspective.
>
> We sincerely thank the reviewer for this crucial question, which allows us to clarify the core value of our method. Following your advice, we **discuss the importance of the LUMoE from the practical and the academic perspective in R3**.
>
> > R6: Q2. I would like to recommend that the authors add the evaluation results on general-purpose benchmarks.
>
> We appreciate the reviewer's suggestion on the additional evaluation. Please refer to **R2 for the additional evaluation results of TruthfulQA and MMBench**.
>
> If you have any further concerns or questions, feel free to let us know. We are glad to discuss and clarify them in more detail.

---

> > ### Author Response · Authors · 2025-11-25
> >
> > Dear reviewer wX2X,
> >
> > Thanks for taking the time to review our work. We have carefully considered your comments and made every effort to respond to your concerns.
> >
> > If you have any further questions or require additional clarification, please let us know.
> >
> > Best regards,
> >
> > Authors

---

> > > ### Comment · Reviewer_wX2X · 2025-11-26
> > > **Response from Reviewer wX2X**
> > >
> > > Thanks to the authors for the dedicated rebuttal.
> > > Accordingly, my concerns regarding W1. the clarification from the related work and W2. the evaluation on general benchmarks have been resolved.
> > >
> > > I still have two remaining concerns.
> > > As a minor concern, the newly added “deep analysis” is not well supported by the experimental results and feels somewhat shallow. It would be helpful to strengthen these analyses so they are more clearly supported by the experimental results.
> > >
> > > My main remaining concern is whether this method truly provides meaningful academic value. I still wonder whether a rather naive approach, introducing a separate LoRA for each request task, provides sufficient novelty.
> > >
> > > For example, as mentioned by Reviewer LUm2, the ICLR2025 paper [2] demonstrates a method that continuously updates a single LoRA by guiding each request to utilize different subspaces within a single LoRA parameter. Compared with such an approach, the method proposed in this work appears limited in academic value, given that LUMoE increases the number of LoRA modules proportionally to the number of requests and even requires an external API for routing.
> > >
> > > Since Reviewer LUm2 also raised this issue, it is clearly a major limitation of the paper.
> > >
> > > Of course, I recognize that the paper does offer a certain contribution from the perspective of benchmark construction (MLUBench, created by automatic crawling from Google Images, answer generation using GPT-4o, and manual filtering). However, the current draft positions LUMoE as the major contribution, and therefore, the novelty of LUMoE becomes a key factor in assessing the paper’s acceptance.
> > >
> > >
> > > Due to the above reasons, I have decided to maintain my score currently.
> > > Regarding this point, I plan to discuss further with the other reviewers and the AC during the upcoming phases.
> > >
> > >
> > > [2] Gao+, On Large Language Model Continual Unlearning, ICLR2025

---

> > > > ### Author Response · Authors · 2025-11-28
> > > >
> > > > # Thanks for your reply!
> > > >
> > > > Dear reviewer wX2X,
> > > >
> > > > We appreciate your engagement with our rebuttal. Now we address your remaining concerns.
> > > >
> > > > > As a minor concern, the newly added “deep analysis” is not well supported by the experimental results and feels somewhat shallow. It would be helpful to strengthen these analyses so they are more clearly supported by the experimental results.
> > > >
> > > > We agree that our initial "Deeper Analysis" is presented as a hypothesis and lacked direct experimental support. Following your advice, we have **conducted new experiments** designed to validate our analysis. We test the GA and KL methods in two scenarios:
> > > > * **Unlearn-LLM-Only:** Freezes the vision components (vision encoder and multimodal projector) and only updates the LLM during lifelong unlearning.
> > > > * **Unlearn-Vision-Only:** Freezes the LLM and only updates the vision components during lifelong unlearning.
> > > >
> > > > The results are presented in the following tables and updated in the **Isolate Experiments** in the Appendix.
> > > >
> > > > ### Table: Results of Unlearn-LLM-Only (LLaVA-7B).
> > > > | Method | Metric | A-UA | A-UB | A-UC | A-UD | B-UB | B-UC | B-UD | C-UC | C-UD | D-UD |
> > > > |--------|----------------|-------|-------|-------|-------|-------|-------|-------|-------|-------|-------|
> > > > | GA | Forget Quality | 0.205 | 0.070 | 0.000 | 0.010 | 0.193 | 0.045 | 0.011 | 0.065 | 0.025 | 0.100 |
> > > > | | Model Utility | 0.102 | 0.023 | 0.000 | 0.000 | 0.308 | 0.050 | 0.016 | 0.000 | 0.000 | 0.000 |
> > > > | KL | Forget Quality | 0.355 | 0.140 | 0.040 | 0.040 | 0.255 | 0.113 | 0.103 | 0.345 | 0.145 | 0.035 |
> > > > | | Model Utility | 0.184 | 0.007 | 0.000 | 0.000 | 0.333 | 0.141 | 0.100 | 0.069 | 0.061 | 0.007 |
> > > >
> > > > ### Table: Results of Unlearn-Vision-Only (LLaVA-7B).
> > > > | Method | Metric | A-UA | A-UB | A-UC | A-UD | B-UB | B-UC | B-UD | C-UC | C-UD | D-UD |
> > > > |--------|----------------|-------|-------|-------|-------|-------|-------|-------|-------|-------|-------|
> > > > | GA | Forget Quality | 0.315 | 0.015 | 0.000 | 0.000 | 0.000 | 0.000 | 0.000 | 0.000 | 0.000 | 0.000 |
> > > > | | Model Utility | 0.246 | 0.046 | 0.000 | 0.000 | 0.017 | 0.000 | 0.000 | 0.007 | 0.007 | 0.000 |
> > > > | KL | Forget Quality | 0.475 | 0.410 | 0.333 | 0.150 | 0.272 | 0.220 | 0.185 | 0.400 | 0.235 | 0.245 |
> > > > | | Model Utility | 0.484 | 0.254 | 0.204 | 0.138 | 0.333 | 0.265 | 0.141 | 0.184 | 0.106 | 0.200 |
> > > >
> > > >
> > > > Our newly added experiments, together with the main experiments in the main text, provide strong and direct evidence for each point in our analysis:
> > > > * **Evidence for LLM-Side Degradation:** The Unlearn-LLM-Only experiments show a sharp decline in performance on our benchmark. This supports our claim that continuously updating the LLM erodes its general capabilities, even without any changes to the vision side.
> > > > * **Evidence for Vision-Side Degradation:** The Unlearn-Vision-Only experiments also result in a significant drop on our benchmark. This confirms our analysis that damage to the vision components also damages MLLMs' general capabilities.
> > > > * **Evidence for Multimodal Alignment Breakdown:** In our main experiments of the main text, we fine-tune both the LLM and the vision components. The severe degradation results of the main experiments and our new experiments confirm that the MLLM is critically dependent on the stable alignment between modalities. Perturbing either side or both sides is sufficient to break this alignment, leading to a complete collapse.

---

> > > > > ### Author Response · Authors · 2025-11-28
> > > > >
> > > > > > My main remaining concern is whether this method truly provides meaningful academic value. I still wonder whether a rather naive approach, introducing a separate LoRA for each request task, provides sufficient novelty.
> > > > >
> > > > > We would like to **articulate the core design principles of our LUMoE and its novelty in the specific context of MLLM lifelong unlearning.** We believe our LUMoE is not a naive simplification but a deliberate and necessary design choice tailored to the MLLMs, which differ from the LLM-only setting considered by [1].
> > > > >
> > > > > ## Response to the Novelty and Design of LUMoE
> > > > >
> > > > > ### 1. Why Separate Adapters, Not a Single Orthogonalized Adapter?
> > > > >
> > > > > The comparison to $O^3$ in [1] is insightful, but we believe their single-LoRA approach **may not be directly transferable to MLLMs.** The core reason is the **increased complexity and parameter update demand of multimodal unlearning.** An MLLM unlearning request requires modifying the model's behavior for both textual and visual inputs. This multi-modality unlearning may need more parameter shifts than a text-only task. A single, shared LoRA module may face a risk of capacity saturation and cross-modal interference when handling a sequence of multimodal requests. Our choice to use separate, isolated adapters is a direct strategy to guarantee capacity and prevent interference in this more demanding environment.
> > > > >
> > > > > ### 2. Practicality and Scalability
> > > > >
> > > > > We understand the concern about a proportional increase in modules. However, we respectfully argue that this concern is insignificant in practice.
> > > > >
> > > > > * **Storage is Trivial:** In our experiments, a single LoRA adapter is about 170MB. Even dozens of adapters would constitute a trivial storage demand on modern systems.
> > > > > * **Not One Adapter Per Entity:** Crucially, our method is more scalable than perceived. As our experiments show, **one adapter can effectively handle the unlearning of a group of entities**, significantly reducing the number of required modules for real-world scenarios.
> > > > >
> > > > > ### 3. The Necessity of a Multimodal Router
> > > > >
> > > > > The reviewer's concern about an external router is understandable. However, for MLLM lifelong unlearning, **a powerful multimodal router is not a flaw but a requirement and strength.** The router must decide when and how to activate an unlearning adapter based on **both the visual and textual input.**
> > > > >
> > > > > * **$O^3$ in [1] Also Needs a Router:** The $O^3$ method is not router-free; it relies on an **OOD detector to decide whether and to what extent to load the unlearning LoRA (Figure 1 in [1]).** Therefore, we believe the fundamental need for a routing mechanism is shared.
> > > > > * **Leveraging Powerful APIs is a Strength, not a Limitation:** Given the **difficulty of multimodal understanding**, leveraging a powerful commercial API as a router is an effective choice. It ensures high-performance routing without the need for training and maintaining a complex multimodal router locally, thus reducing local computational burdens.
> > > > >
> > > > > ### 4. The Novelty of LUMoE
> > > > >
> > > > > We would like to rearticulate the algorithmic and architectural novelty of LUMoE. While LoRA and MoE are established techniques, to the best of our knowledge, LUMoE is the first attempt to successfully synthesise them into a coherent solution for the challenging problem of lifelong unlearning for MLLMs. Therefore, LUMoE's contribution is in proposing an effective architectural paradigm for the MLLM lifelong unlearning.
> > > > >
> > > > > ## Response to Contribution of Our Paper
> > > > >
> > > > > Regarding the broader point about our paper's contribution, we appreciate the reviewer acknowledging the value of our benchmark MLUBench. We would like to gently emphasize that our paper was submitted to the **"Datasets and Benchmarks" track.**
> > > > >
> > > > > In this context, we believe our method LUMoE and the benchmark MLUBench should be viewed as **an integrated contribution rather than isolated contributions.** Specifically, MLUBench enables a rigorous evaluation of the MLLM Lifelong Unlearning problem, while LUMoE serves as an effective method to address the challenges introduced by MLLM Lifelong Unlearning. Together, they form **a cohesive contribution by introducing both a broad-scale benchmark and an effective solution.**
> > > > >
> > > > > [1] Gao+, On Large Language Model Continual Unlearning, ICLR2025
> > > > >
> > > > > We hope our clarifications and revisions can address your remaining concerns.
> > > > >
> > > > > Best regards,
> > > > >
> > > > > Authors

---

### Official Review · Reviewer_LUm2 · 2025-10-31

**Soundness:** 2
**Presentation:** 2
**Contribution:** 3
**Rating:** 4
**Confidence:** 3

**Summary:**

This paper focuses on sequential (lifelong) unlearning for multimodal large language models (MLLMs). The main contributions claimed in this paper are:

- MLUBench: A new benchmark, which consists of 127 entities across 9 domains, including 5,105 images and 15,414 visual question–answer (VQA) pairs, designed to evaluate the degradation and accumulation effects that occur in continual unlearning scenarios.

- LUMoE: A sequential unlearning method for MLLM based on LoRA-based Mixture-of-Experts (MoE). Each unlearning task is handled by a dedicated LoRA adapter, and a routing (gate) module dynamically assigns the appropriate adapter at inference time. This design enables the model to forget specific knowledge continuously without modifying the base model.

Experimental results on LLaVA-1.6-7B and 13B demonstrate that conventional unlearning methods (e.g., GA, GD, KL, NPO) suffer from severe cumulative forgetting and language degradation, whereas LUMoE maintains both high forget quality and model utility across all sequential unlearning steps.

**Strengths:**

**S1.** Sequential unlearning is a highly important yet challenging problem, representing one of the key open issues in current machine unlearning research. It is valuable that this paper explicitly focuses on this problem in the context of MLLMs.

**S2.** MLUBench is one of the largest benchmarks for MLLM unlearning, in terms of the number of images and VQA pairs.

**S3.** LUMoE demonstrates that a MoE architecture is highly effective for sequential unlearning in MLLMs, similar to recent findings in sequential unlearning for LLMs and other model architectures.

**S4.** Experimental results show consistently strong performance, significantly outperforming existing unlearning baselines.

**Weaknesses:**

**W1. Novelty of MLLM Lifelong Unlearning**

This paper claims, as one of its contributions, to define a new problem, "MLLM Lifelong Unlearning." While the problem is indeed important and challenging, a highly similar setting has already been introduced in MUSE (Shi et al., 2024) and [a]. Although the authors cite MUSE, the paper does not explicitly acknowledge or discuss this overlap.

These prior works focus on LLMs rather than MLLMs, but the conceptual difference appears minor. Moreover, they already demonstrated that existing unlearning methods such as GA, NPO, and their variants fail in sequential unlearning settings, which is the same conclusion presented in this paper.

This paper should explicitly describe existing attempts on sequential unlearning, including MUSE and [a]. Without such clarification, the claimed novelty becomes ambiguous and may undermine the credibility of the work.


**W2. Novelty of LUMoE**

From a technical perspective, the novelty of LUMoE is rather limited. LoRA-based adapters with routing mechanisms have already been used for sequential unlearning in LLMs [a]. Furthermore, the use of MoE architectures for continual learning or lifelong model editing is well established (e.g., [b]). The key components (e.g., LoRA, learning algorithm, and GLM-4V-Plus) are all existing techniques. Consequently, the contribution of LUMoE lies more in engineering integration than in algorithmic novelty.


**W3. Method Design**

While LUMoE effectively handles sequential unlearning by isolating LoRA adapters per entity, the current routing design activates only a single adapter for each input, as the paper notes "If a request matches multiple existing tasks, any of the corresponding adapters can be routed into the base model." This implies that when multiple unlearned entities co-occur in a single query, the model may still output information associated with the non-selected adapter(s), leading to partial forgetting and potential leakage.


[a] Gao et al., On Large Language Model Continual Unlearning, ICLR 2025.

[b] Wang & Li, LEMoE: Advanced Mixture of Experts Adaptor for Lifelong Model Editing of Large Language Models, EMNLP 2024.

**Questions:**

**W1.** This is the most critical issue and should be resolved before the paper can be accepted.

**W2.** Please correct me if there is any misunderstanding.

**W3.** When multiple forgotten entities co-occur in a query, do the authors have any mitigation strategy or evidence showing that partial forgetting or leakage does not occur? Alternatively, is there any reasonable solution to address this problem?

---

> ### Author Response · Authors · 2025-11-20
> **Response to reviewer LUm2**
>
> # Response to reviewer LUm2
>
> We are glad the reviewer found the studied problem highly important. We would like to first address the reviewer's most critical concern, then respond to each point individually.
>
> > R1: W1. Novelty of MLLM Lifelong Unlearning
>
> We really appreciate your suggestions on the discussion of these two outstanding works [1,2]. Following your advice, we **have cited and thoroughly discussed** them and **acknowledged** their **contributions** in the **related work** of our submission, as shown below.
>
> **Sequential Unlearning of Language Models**
>
> The growing body of work on sequential unlearning for Large Language Models (LLMs) provides a critical foundation for our proposed MLLM Lifelong Unlearning. Previous studies have grappled with core challenges in this area. Gao et al. [1], for example, tackled the trade-off between unlearning efficacy and model utility, introducing the $O^3$ framework to navigate this balance without relying on retained data. In a complementary study, Shi et al. [2] evaluated the sustainability of unlearning methods, determining that they are ill-equipped for sequential unlearning requests. This conclusion resonates with our own analysis, underscoring that continual unlearning remains a significant and persistent challenge across both LLMs and MLLMs.
>
> [1] Gao et al., On Large Language Model Continual Unlearning, ICLR 2025.
>
> [2] Shi W, Lee J, Huang Y, et al. MUSE: Machine Unlearning Six-Way Evaluation for Language Models[C]//The Thirteenth International Conference on Learning Representations.
>
> > R2: W2. Novelty of LUMoE and Question 2
>
> Thank you for this insightful comment. We agree that the individual components, like LoRA and MoE, are established. However, we respectfully argue that the novelty of LUMoE lies not in inventing new components, but in their **novel integration and adaptation to solve the uniquely challenging problem of MLLM lifelong unlearning**.
>
> This presents three key challenges that existing methods do not fully address:
>
> - **Multimodal Catastrophic Forgetting**: Unlike text-only models, unlearning in MLLMs must prevent degradation across both visual and linguistic modalities. LUMoE's design explicitly manages this cross-modal interference.
> - **Complex Multimodal Associations**: A single concept (e.g., "Eiffel Tower") exists as an image, a text description, and complex cross-modal embeddings. Our MoE-based routing mechanism is novel in its application to identify and isolate these rich, multimodal concepts for targeted unlearning.
> - **Scalability for Lifelong MLLM unlearning**: While MoE is known for scalability, applying it to the lifelong unlearning setting for massive MLLMs is an unexplored task. Our work provides the first demonstrated framework for efficient, scalable unlearning in this context.
>
> Therefore, LUMoE's contribution is algorithmic and architectural: it is a novel framework that re-purposes and significantly adapts existing tools (MoE, LoRA) to create an effective solution for a new and important problem. We believe this represents a meaningful step forward for the community.
>
>
> To better clarify our contributions towards prior works, we involved and discussed more references in the **related work** of our submission. For instance, while [3] uses MoE for lifelong model editing, our router is specifically designed to handle multimodal keys (visual and textual features). Similarly, while [4] uses MoE for continual learning, its application to the unlearning objective in MLLMs, with our proposed novel framework and benchmark, is a distinct contribution.
>
> [3] Wang & Li, LEMoE: Advanced Mixture of Experts Adaptor for Lifelong Model Editing of Large Language Models, EMNLP 2024.
>
> [4] Rypeść G, Cygert S, Khan V, et al. Divide and not forget: Ensemble of selectively trained experts in continual learning[J]. arXiv preprint arXiv:2401.10191, 2024.
>
> > R3: W3. Method Design and Question 3
>
> Thank the reviewer for raising this critical point. We would like to clarify our method design and its robustness in this case.
>
> Our current design of activating a single adapter at a time can handle most cases of a single forgotten entity. We believe these cases are common in real-world scenarios. Thus, this design ensures efficiency.
>
> **For the multi-entity case the reviewer described**, our modular LUMoE framework can **provide a simple yet effective solution: merging of multiple adapters**. Specifically, when the gate module detects multiple entities, we can merge all their corresponding adapters into the base model. The LoRA supports such action. Then, the combined effects of these adapters can effectively prevent the partial leakage.
>
> If you have any further concerns or questions, feel free to let us know. We are glad to discuss and clarify them in more detail.

---

> > ### Author Response · Authors · 2025-11-25
> >
> > Dear reviewer LUm2,
> >
> > Thanks for taking the time to review our work. We have carefully considered your comments and made every effort to respond to your concerns.
> >
> > If you have any further questions or require additional clarification, please let us know.
> >
> > Best regards,
> >
> > Authors

---

> > > ### Comment · Reviewer_LUm2 · 2025-11-26
> > >
> > > I appreciate the authors’ efforts to address my questions. However, I believe further clarification is needed regarding W1 and W3.
> > >
> > > **W1. Novelty of MLLM Lifelong Unlearning**
> > >
> > > I remain unsure how the proposed MLLM Lifelong Unlearning fundamentally differs from prior work on sequential unlearning such as MUSE [1], O3 [2], or PULSE [3]. In light of [1] and [2], is MLLM Lifelong Unlearning essentially a straightforward extension of the same idea from LLMs to MLLMs? Furthermore, from the perspective of problem definition, how does it differ from [3]?
> > >
> > > Because the paper positions the formulation of MLLM Lifelong Unlearning itself as a core contribution, its distinction from existing sequential unlearning research should be articulated clearly and should constitute a meaningful conceptual innovation. Without such clarification, the novelty of the problem setting remains ambiguous.
> > >
> > > **W3. Method Design**
> > >
> > > The authors claim that multi-entity unlearning can be handled by merging multiple LoRA modules. I had also considered this possibility, but I remain unconvinced. This approach implicitly assumes that LoRA modules trained for different unlearning requests *do not interfere with one another* and *are additively composable*, yet the paper includes no mechanism to ensure such properties.
> > >
> > > O3 [1] explicitly enforces orthogonality during LoRA updates to avoid interference, which in turn enables implicit routing across different unlearning directions. In contrast, the proposed LUMoE includes no such mechanism, and it is unclear whether MoE routing alone can guarantee the required behavior.
> > >
> > >
> > > Other then these two, I also have similar idea as Reviewer wX2X about the novelty of LUMoE (my W2).
> > >
> > >
> > > [1] Gao et al., On Large Language Model Continual Unlearning, ICLR 2025.
> > >
> > > [2] Shi et al., MUSE: Machine Unlearning Six-Way Evaluation for Language Models, ICLR 2025.
> > >
> > > [3] Kawakami et al., PULSE: Practical Evaluation Scenarios for Large Multimodal Model Unlearning, NeurIPS Workshop 2025.

---

> > > > ### Author Response · Authors · 2025-11-28
> > > >
> > > > # Thanks for your reply!
> > > >
> > > > Dear reviewer LUm2,
> > > >
> > > > We appreciate your engagement with our rebuttal. Now we address your remaining concerns.
> > > >
> > > > > W1. Novelty of MLLM Lifelong UnlearningI remain unsure how the proposed MLLM Lifelong Unlearning fundamentally differs from prior work on sequential unlearning such as MUSE [1], O3 [2], or PULSE [3]. In light of [1] and [2], is MLLM Lifelong Unlearning essentially a straightforward extension of the same idea from LLMs to MLLMs? Furthermore, from the perspective of problem definition, how does it differ from [3]?Because the paper positions the formulation of MLLM Lifelong Unlearning itself as a core contribution, its distinction from existing sequential unlearning research should be articulated clearly and should constitute a meaningful conceptual innovation. Without such clarification, the novelty of the problem setting remains ambiguous.
> > > >
> > > > Thank the reviewer for raising this critical point. We would like to first clarify that MLLM lifelong unlearning **is not a straightforward extension of the same idea from LLMs to MLLMs**, but a distinct concept and **more challenging problem**. Specifically, we believe the core distinction lies in the **multimodal alignment**, which introduces a unique challenge not present in unimodal LLMs.
> > > >
> > > > The challenge of MLLM lifelong unlearning is that methods have to preserve the integrity of both the Language Model and the vision components (vision adapter and multimodal projector), and the alignment that bridges them.
> > > >
> > > > To empirically prove our argument, we **conduct new experiments** where we isolate the unlearning process to **update either the language or vision part of MLLMs**. We apply the GA and KL methods under two conditions:
> > > > * **Unlearn-LLM-Only**: We freeze the vision components (vision encoder and multimodal projector) and only update the LLM weights.
> > > > * **Unlearn-Vision-Only**: We freeze the LLM and only update the vision encoder and multimodal projector.
> > > >
> > > > The results are in the following tables.
> > > >
> > > > ### Table: Results of Unlearn-LLM-Only (LLaVA-7B).
> > > > | Method | Metric | A-UA | A-UB | A-UC | A-UD | B-UB | B-UC | B-UD | C-UC | C-UD | D-UD |
> > > > |--------|----------------|-------|-------|-------|-------|-------|-------|-------|-------|-------|-------|
> > > > | GA | Forget Quality | 0.205 | 0.070 | 0.000 | 0.010 | 0.193 | 0.045 | 0.011 | 0.065 | 0.025 | 0.100 |
> > > > | | Model Utility | 0.102 | 0.023 | 0.000 | 0.000 | 0.308 | 0.050 | 0.016 | 0.000 | 0.000 | 0.000 |
> > > > | KL | Forget Quality | 0.355 | 0.140 | 0.040 | 0.040 | 0.255 | 0.113 | 0.103 | 0.345 | 0.145 | 0.035 |
> > > > | | Model Utility | 0.184 | 0.007 | 0.000 | 0.000 | 0.333 | 0.141 | 0.100 | 0.069 | 0.061 | 0.007 |
> > > >
> > > > ### Table: Results of Unlearn-Vision-Only (LLaVA-7B).
> > > > | Method | Metric | A-UA | A-UB | A-UC | A-UD | B-UB | B-UC | B-UD | C-UC | C-UD | D-UD |
> > > > |--------|----------------|-------|-------|-------|-------|-------|-------|-------|-------|-------|-------|
> > > > | GA | Forget Quality | 0.315 | 0.015 | 0.000 | 0.000 | 0.000 | 0.000 | 0.000 | 0.000 | 0.000 | 0.000 |
> > > > | | Model Utility | 0.246 | 0.046 | 0.000 | 0.000 | 0.017 | 0.000 | 0.000 | 0.007 | 0.007 | 0.000 |
> > > > | KL | Forget Quality | 0.475 | 0.410 | 0.333 | 0.150 | 0.272 | 0.220 | 0.185 | 0.400 | 0.235 | 0.245 |
> > > > | | Model Utility | 0.484 | 0.254 | 0.204 | 0.138 | 0.333 | 0.265 | 0.141 | 0.184 | 0.106 | 0.200 |
> > > >
> > > > According to the results, **in both scenarios, the model's overall performance suffers severe, cumulative degradation.** The results are similar to our main experiments, which unlearn both the LLM and vision modules.
> > > >
> > > > Our new results, alongside our experiments in the main text, demonstrate that **MLLM lifelong unlearning is not a problem that can be solved by addressing one modality in isolation**. The MLLMs' overall ability is dependent on the stable alignment between modalities. Unlearning methods that perturb the weights of either LLM or vision components risk catastrophically breaking this alignment.
> > > >
> > > > We have updated the above discussion in our **problem formulation** and the detailed results in the **Isolate Experiments** in the Appendix.
> > > >
> > > > Now we **discuss the difference between our definition and the definition in [3]**. PULSE [3] provides an excellent evaluative framework by defining the "Long-term Sustainability" scenario. Our definition further identifies the fundamental challenge within that scenario. Specifically, PULSE identifies the potential breakdown of alignment as an interesting phenomenon but explicitly **leaves its rigorous investigation as "future work" (Section 4.4 of PULSE).** **Our definition directly investigates this** through targeted experiments of isolating both the LLM and vision parts. Therefore, we offer a deeper and more explicit problem definition.

---

> > > > > ### Author Response · Authors · 2025-11-28
> > > > >
> > > > > > W3. Method DesignThe authors claim that multi-entity unlearning can be handled by merging multiple LoRA modules. I had also considered this possibility, but I remain unconvinced. This approach implicitly assumes that LoRA modules trained for different unlearning requests do not interfere with one another and are additively composable, yet the paper includes no mechanism to ensure such properties.O3 [1] explicitly enforces orthogonality during LoRA updates to avoid interference, which in turn enables implicit routing across different unlearning directions. In contrast, the proposed LUMoE includes no such mechanism, and it is unclear whether MoE routing alone can guarantee the required behavior.
> > > > >
> > > > > We agree that additively merging LoRA modules trained for different tasks may lead to destructive interference. However, our method is designed to train experts to perform **a unified refusal behavior, which naturally mitigates this issue.**
> > > > >
> > > > > To empirically validate this, we **conduct additional experiments.** Specifically, we train five separate refusal adapters for five sequential unlearning tasks (A, B, C, D, E) discussed in the "Impact of number of tasks" in Section 6.4. Then we progressively merge them (e.g., A+B, A+B+C). After each merge, we tested the model's unlearning quality on all unlearned tasks. Our results show that the merge of multiple adapters has **no significant interference**.
> > > > >
> > > > > The table shows the Forget Quality on each task after each merge. The "Individual Adapter" row serves as the baseline, showing the performance of each adapter on its specific task without any merging.
> > > > >
> > > > > ### Table: Interference Assessment of Merged Refusal Adapters.
> > > > > | Merged Adapters | Task A | Task B | Task C | Task D | Task E |
> > > > > |:---------------------|:------:|:------:|:------:|:------:|:------:|
> > > > > | Individual Adapter | 1.00 | 0.95 | 0.99 | 0.96 | 1.00 |
> > > > > | A + B | 1.00 | 1.00 | - | - | - |
> > > > > | A + B + C | 1.00 | 1.00 | 1.00 | - | - |
> > > > > | A + B + C + D | 1.00 | 1.00 | 1.00 | 1.00 | - |
> > > > > | A + B + C + D + E | 1.00 | 1.00 | 1.00 | 1.00 | 1.00 |
> > > > >
> > > > > As shown in the table, the Forget Quality on each task after merging **even surpasses the individual adapter.** This confirms that our additive merging approach for refusal adapters does not introduce destructive interference.
> > > > >
> > > > > We have updated the additional experiments in the **"Interference Assessment"** section in the Appendix.
> > > > >
> > > > > We also provide an **intuitive explanation** for this non-interference phenomenon. We believe that, unlike standard fine-tuning, where LoRA modules learn to output different, potentially conflicting facts (e.g., Task A: "Answer is X," Task B: "Answer is Y"), **our adapters all learn the same refusal behavior.**

---

> > > > > > ### Author Response · Authors · 2025-11-28
> > > > > >
> > > > > > > Other then these two, I also have similar idea as Reviewer wX2X about the novelty of LUMoE (my W2).
> > > > > >
> > > > > > ## Response to the Novelty and Design of LUMoE
> > > > > >
> > > > > > ### 1. Why Separate Adapters, Not a Single Orthogonalized Adapter?
> > > > > >
> > > > > > The comparison to $O^3$ in [1] is insightful, but we believe their single-LoRA approach **may not be directly transferable to MLLMs.** The core reason is the **increased complexity and parameter update demand of multimodal unlearning.** An MLLM unlearning request requires modifying the model's behavior for both textual and visual inputs. This multi-modality unlearning may need more parameter shifts than a text-only task. A single, shared LoRA module may face a risk of capacity saturation and cross-modal interference when handling a sequence of multimodal requests. Our choice to use separate, isolated adapters is a direct strategy to guarantee capacity and prevent interference in this more demanding environment.
> > > > > >
> > > > > > ### 2. Practicality and Scalability
> > > > > >
> > > > > > We understand the concern about a proportional increase in modules. However, we respectfully argue that this concern is insignificant in practice.
> > > > > >
> > > > > > * **Storage is Trivial:** In our experiments, a single LoRA adapter is about 170MB. Even dozens of adapters would constitute a trivial storage demand on modern systems.
> > > > > > * **Not One Adapter Per Entity:** Crucially, our method is more scalable than perceived. As our experiments show, **one adapter can effectively handle the unlearning of a group of entities**, significantly reducing the number of required modules for real-world scenarios.
> > > > > >
> > > > > > ### 3. The Necessity of a Multimodal Router
> > > > > >
> > > > > > The reviewer's concern about an external router is understandable. However, for MLLM lifelong unlearning, **a powerful multimodal router is not a flaw but a requirement and strength.** The router must decide when and how to activate an unlearning adapter based on **both the visual and textual input.**
> > > > > >
> > > > > > * **$O^3$ in [1] Also Needs a Router:** The $O^3$ method is not router-free; it relies on an **OOD detector to decide whether and to what extent to load the unlearning LoRA (Figure 1 in [1]).** Therefore, we believe the fundamental need for a routing mechanism is shared.
> > > > > > * **Leveraging Powerful APIs is a Strength, not a Limitation:** Given the **difficulty of multimodal understanding**, leveraging a powerful commercial API as a router is an effective choice. It ensures high-performance routing without the need for training and maintaining a complex multimodal router locally, thus reducing local computational burdens.
> > > > > >
> > > > > > ### 4. The Novelty of LUMoE
> > > > > >
> > > > > > We would like to rearticulate the algorithmic and architectural novelty of LUMoE. While LoRA and MoE are established techniques, to the best of our knowledge, LUMoE is the first attempt to successfully synthesise them into a coherent solution for the challenging problem of lifelong unlearning for MLLMs. Therefore, LUMoE's contribution is in proposing an effective architectural paradigm for the MLLM lifelong unlearning.
> > > > > >
> > > > > > [1] Gao+, On Large Language Model Continual Unlearning, ICLR2025
> > > > > >
> > > > > > [3] PULSE: Practical Evaluation Scenarios for Large Multimodal Model Unlearning, NeurIPS Workshop 2025
> > > > > >
> > > > > > We hope our clarifications and revisions can address your remaining concerns.
> > > > > >
> > > > > > Best regards,
> > > > > >
> > > > > > Authors

---

### Meta-Review · Area_Chair_PAuc · 2026-01-08

**Summary:**

This paper studies sequential (lifelong) unlearning for multimodal LLMs, introducing the MLUBench benchmark and proposing LUMoE, a LoRA-based MoE approach that routes unlearning requests to task-specific refusal adapters. Experiments on LLaVA show that standard unlearning baselines degrade under sequential requests, while LUMoE better maintains forget quality and utility.

The main concern across reviewers is novelty and positioning: sequential/lifelong unlearning has been studied for LLMs and related multimodal unlearning settings exist, so the “MLLM lifelong unlearning” framing and LUMoE are viewed as relatively straightforward extensions and largely engineering-driven. During rebuttal, the authors improved related-work coverage and added additional evaluations, addressing several presentation and evaluation concerns. However, the core novelty concern remains, and reviewers were not fully convinced that the conceptual contribution and algorithmic novelty sufficiently exceed prior sequential unlearning approaches. Given these concerns, the AC recommends the paper below the acceptance line.

**Reviewer Concerns:**

Concerns that were substantially addressed during rebuttal:
- Related work coverage: Reviewers’ requests to better acknowledge and discuss the work relative to existing sequential unlearning and multimodal unlearning literature were addressed with expanded discussion and citations.
- General-purpose utility evaluation: The authors added evaluations on general-purpose benchmarks to support the claim that utility is preserved.
- Generalizability beyond LLaVA and additional ablations: Additional pre-unlearning accuracy checks on other MLLMs and expanded ablation/analysis were provided.

Outstanding concern driving the decision:
- Core novelty remains unconvincing: Despite clarifications and new “LLM-only vs. vision-only” experiments, it remained unconvincing that the main claimed contribution, the novelty of the lifelong MLLM unlearning problem definition and the proposed solution paradigm, is sufficiently distinct from existing sequential unlearning and multimodal unlearning lines of work. In particular, the revised paper itself still frames and emphasizes “lifelong” as the primary novelty, while the multimodal aspect appears more like a natural extension of established settings and methods.

**Reviewer Scores:**

All three reviewers participated in the early-stage discussion. Given the remaining novelty concerns, Reviewers LUm2 and wX2X are unlikely to increase their scores. Reviewer cC9S had most of their concerns addressed during rebuttal and is the reviewer most likely to increase their score.

---

### Decision · Program_Chairs · 2026-01-26

Reject